# Associated factors of health-related quality of life in Indonesian Women with Systemic Lupus Erythematosus: A cross-sectional within-cohort analysis

Euphemia Seto Anggraini Widyastuti[1]◐¤, Josephine Lavina[1‡], Gabriel Justin Darmajaya[1‡], Stevent Sumantri [1,2]*◐¤

1 Department of Internal Medicine, Faculty of Medicine, Universitas Pelita Harapan, Tangerang, Banten, Indonesia, 2 Allergy and Clinical Immunology Division, Department of Internal Medicine, Faculty of Medicine, Universitas Pelita Harapan, Tangerang, Banten, Indonesia

¤ Current Address: Boulevard Jenderal Sudirman, Department of Internal Medicine, Faculty of Medicine, Universitas Pelita Harapan, Lippo Village, Tangerang, Banten, Indonesia
◐ These authors contributed equally to this work.
‡ These authors also contributed equally to this work.
* stevent.sumantri@uph.edu

## Abstract

### Objective

This study aims to determine the effects of sleep quality along with age, marital status, socioeconomic status, depression, anxiety, disease activity, pain scale, and dose of corticosteroids on quality of life in women with SLE.

### Methods

Variables were assessed in 75 women with SLE using the Pittsburgh Sleep Quality Index (PSQI), Lupus Quality of Life (Lupus QoL), Depression, Anxiety, and Stress Scale-21 (DASS-21), and Mexican SLE Disease Activity Index (MEX-SLEDAI). Bivariate and multivariate analyses were performed to determine contributors to quality of life.

### Results

Of 75 subjects, 35 (46.7%) patients had poor sleep quality. The mean QoL score for patients is 84.27. Poor sleepers had impaired QoL in physical health (p = 0.003), emotional health (p = 0.007), pain (p = 0.003), and planning (p = 0.006), with fatigue (p < 0.0001) as the most significantly impaired. Younger age (Mean ± SD = 81.1 ± 12.67; p = 0.014) and anxiety or depression (Mean ± SD = 56.66 ± 8.17; p = 0.006) were significantly associated with lower quality-of-life scores. The linear regression results showed an R-squared of 0.361, with anxiety (β = 21.402), sleep quality (β = 8.392), and

**Data availability statement:** All relevant data are within the manuscript and its Supporting Information files.

**Funding:** The author(s) received no specific funding for this work.

age ($\beta = 5.526$) as the most significant variables. Marital status, socioeconomic status, disease activity, pain scale, and corticosteroid dose did not correlate with QoL.

## Conclusion

Poor sleep quality, anxiety, and younger age were significant independent predictors of lower QoL in women with SLE, explaining 36.1% of the variance. These findings suggest that psychosocial and sleep interventions are crucial for improving well-being in this population, potentially more so than focusing solely on disease activity.

## Introduction

While high-impact, peer-reviewed studies detailing the precise nationwide incidence and prevalence of Systemic Lupus Erythematosus (SLE) in Indonesia are still emerging, a comprehensive 2021 systematic review highlights that crude incidence rates in the region range from 0.9 to 3.1 per 100,000 persons per year, with prevalence rates between 4.3 and 45.3 per 100,000 persons. The study also confirms that Asian populations, including those in Southeast Asia, often experience higher rates of severe organ involvement, particularly lupus nephritis, compared to Caucasian populations, underscoring the significant health burden of SLE in Indonesia and neighboring countries [1]. Poor sleep quality, prevalent in approximately 62% of SLE patients, is a critical factor contributing to diminished QoL, compounded by psychological factors such as depression and anxiety, as well as disease activity, pain, and corticosteroid use [2].

Despite its importance, the relationship between sleep quality and QoL among patients with SLE remains underexplored, particularly in the Indonesian context [2]. The experience of an SLE patient in Indonesia is uniquely shaped by a genetic predisposition to more severe disease [3], cultural norms that can stigmatize mental health and influence treatment choices, and significant socioeconomic pressures that affect access to and adherence with care [4]. These factors combined create a complex web where the relationships between sleep, mental health, and QoL are intensified compared to other populations.

This study aims to investigate the association between sleep quality and QoL among female SLE patients, while also examining the influence of age, socioeconomic status, depression, anxiety, disease activity, pain, and corticosteroid use. We hypothesized that poor sleep quality would be independently associated with lower QoL in female SLE patients, even after controlling for disease activity, pain, and demographic factors.

Although previous studies have established that patients with SLE generally experience lower health-related quality of life (HRQoL) than healthy populations, the relative importance of modifiable factors, particularly sleep quality and psychosocial health, in the Indonesian SLE population remains unclear. This study was therefore designed as a within-cohort analysis to identify the primary drivers of HRQoL variance. By focusing exclusively on this patient population, the study seeks to delineate

the specific impact of sleep disturbances and psychiatric symptoms on daily functioning, independent of systemic disease activity.

## Materials and methods

### Patients and settings

A cross-sectional, within-cohort study was conducted among women with SLE who sought treatment at the Allergy-Clinical Immunology Outpatient Clinic at Siloam Hospital Lippo Village. This design was specifically chosen to evaluate internal predictors of HRQoL in patients with established SLE.

Data were collected from February to May 2023 using a questionnaire distributed to all subjects who met the inclusion and exclusion criteria. All female patients with SLE, as defined by the EULAR/ACR 2019 classification criteria and able to communicate clearly, were included consecutively, while women with other autoimmune diseases were excluded.

The study received approval from the Medical Research Ethics Committee, Faculty of Medicine, Universitas Pelita Harapan, Tangerang (190/K-LKJ/ETIK/XI/2022). All subjects received an adequate explanation and provided written informed consent to participate. The reporting of this study conforms to the STROBE (Strengthening the Reporting of Observational Studies in Epidemiology) statement [5].

### Data collection and assessment

In this study, we conducted a post hoc power analysis based on the observed effect size, sample size, and significance level, yielding a power of 0.957. A cutoff of greater than 0.8 is considered appropriate for analysis. We acknowledge that although the power was sufficient for primary analyses, the sample size may limit the ability to detect smaller effects for secondary variables. This limitation affects our interpretation of results concerning variables such as educational status, marital status, and monthly income, which may be underpowered due to our modest sample size. As such, caution should be exercised when generalizing findings regarding these secondary variables, and additional research with larger samples may be needed to elucidate their effects fully.

The variables included were age (young adults, < 40 years; middle-aged adults, 40–59 years; and older adults, > 60 years), in accordance with the Indonesian Ministry of Health criteria, and marital status (single, married, divorced). Socioeconomic status was categorized based on the regional minimum wage (Upah Minimum Provinsi/UMP) of Jakarta/Tangerang for 2023 (approximately IDR 4,901,798). Monthly income was subsequently classified as 'Above Minimum Wage' or 'Below Minimum Wage' to reflect local purchasing power parity. Educational status was defined as 'Low' (primary school or less), 'Middle' (secondary/high school), or 'High' (tertiary/university degree).

Lupus clinical characteristics defined by duration of SLE (months; numeric), dose of corticosteroids (high >7.5 mg/day or low <=7.5 mg/day, prednisone-equivalent dose), QoL (numeric based on Indonesian language LupusQoL) [6], disease activity (Indonesian language MEX-SLEDAI, low 0–4, medium 5–9, high >=10) [7], and anxiety/depression status (Indonesian language DASS-21, yes or no) [8]. For clinical interpretability, variables were dichotomized based on established clinically significant thresholds: Sleep quality was categorized using the validated Indonesian language PSQI [9] cutoff of >5 to distinguish 'poor' from 'good' sleepers. Similarly, pain intensity on the Visual Analog Scale (VAS) was categorized as 'mild' (0–39 mm) or 'severe' (40–100 mm), based on previous HRQoL studies in SLE, which suggest that scores above 40 mm represent a distinct threshold for functional impairment [10].

The Mexican SLE Disease Activity Index (MEX-SLEDAI) was selected as the primary measure of disease activity due to its validated clinical utility in resource-limited settings. Unlike the standard SLEDAI-2K, the MEX-SLEDAI enables accurate assessment through clinical evaluation and basic laboratory parameters without requiring specialized immunological assays (e.g., anti-dsDNA titers or complement levels), which was not accesible in the study site. This makes it a highly practical and reliable tool for cross-sectional studies in diverse Southeast Asian patient populations. However, the

 

MEX-SLEDAI has certain limitations: it may be less sensitive to mild flares and lacks a comprehensive assessment of permanent organ damage. These aspects could affect its ability to capture the full spectrum of disease activity and may introduce variability in comparisons with other indices.

## Statistical analysis

For statistical analysis, IBM SPSS Statistics (version 25) was used to analyze the collected data. We performed the Kolmogorov-Smirnov test and generated a p-value < 0.05, indicating an abnormal data distribution. Therefore, this study employed the Mann-Whitney U and Kruskal-Wallis tests for bivariate analysis. Then, multivariate analysis with linear regressions on variables that met the requirement (p < 0.05) was performed to identify which variables independently influenced subjects' QoL. Multicollinearity among the independent variables was assessed using the Variance Inflation Factor (VIF) and tolerance statistics. All VIF values were < 1.5, indicating that no significant multicollinearity was present in the final regression model.

## Results

A total of 75 subjects were included in this study (Table 1). The majority of the subjects were aged <60 years (96%; n = 72), married (64%; n = 48), had a high education level (50.7%; n = 38), an income above minimum wage (77.3%; n = 58), and had sufficient funds for one month's need (89.3%; n = 67). Regarding the disease, the duration of SLE since diagnosis was 65.4 months or 5.45 years; it was maintained with low-dose steroid therapy (< 7.5 mg/day) in 76% (n = 57); had mild disease activity in 82.7% (n = 62); and had a mild pain score in 61.3% (n = 46). Based on the questionnaires, subjects mostly had good sleep quality (53.3%; n = 40), a mean quality-of-life score of 84.27 ± 12.59, and no depression or anxiety (96%; n = 72).

Major characteristics of the poor sleepers described in this study were young (18–39 years old), rated their sleep quality as "quite poor" subjectively (57.1%; n = 20), had sleep latency >60 minutes (51.4%; n = 18), slept for 6–7 hours (42.9%; n = 15), had poor sleep efficiency (34.3%; n = 12), experienced sleep disturbances <1 time a week (62.9%; n = 22), took sleeping pills <1 time a week (62.9%; n = 22), and experienced daytime dysfunction such as drowsiness and impaired focus >= 3 times a week (40%; n = 14).

### 3.1 Analysis of the association between QoL and SLE-related variables

The bivariate analysis, as presented in Table 2, demonstrated a significant association between sleep quality and quality of life (QoL) among the 75 female with SLE, with patients exhibiting poor sleep quality (n = 35) having a lower mean QoL score of 78.91 ± 13.67 compared to those with good sleep quality (n = 40) at 88.95 ± 9.47 (p < 0.0001). Age showed a notable trend, with younger patients (n = 43) having a mean QoL score of 81.1 ± 12.67, middle-aged patients (n = 29) at 88.13 ± 11.67, and elderly patients (n = 3) at 92.39 ± 7.81 (p = 0.014). Education level also influenced QoL, with patients having elementary education (n = 7) scoring 79.84 ± 18.95, those with middle education (n = 38) scoring 81.71 ± 12.42, and those with high education (n = 30) scoring 88.49 ± 10.07 (p = 0.05). Depression and anxiety, assessed via DASS-21, significantly impacted QoL, with patients experiencing mild to severe depression (n = 3) and depression or anxiety (n = 3), both scoring 56.66 ± 8.17 compared to those with normal levels (n = 72) at 85.42 ± 11.4 (p = 0.006 for both). However, no significant associations were found between QoL and marital status (p = 0.129), monthly income (p = 0.139), sufficiency for one month (p = 0.482), duration of SLE (p = 0.151), use of steroids (p = 0.995), disease activity via MEX-SLEDAI (p = 0.297), or pain score via VAS (p = 0.663), despite slight variations in mean QoL scores across these categories.

### 3.2 Analysis of the association between sleep quality and each domain of quality of life based on LupusQoL

The analysis of the association between sleep quality and each domain of quality of life (QoL) using the LupusQoL questionnaire, as presented in Table 3, revealed significant differences among 75 female patients with Systemic Lupus

**Table 1. Demographic characteristics of participants.**

| Variables | | n(%) |
|---|---|---|
| Age | | |
| | <40 years | 43 (57.3%) |
| | 40-59 years | 29 (38.6%) |
| | ≥60 years | 3 (4.1%) |
| Education Levels | | |
| | Elementary | 7 (9.3%) |
| | Middle (high school) | 38 (50.7%) |
| | High | 30 (40%) |
| Marital Status | | |
| | Married | 48 (64%) |
| | Unmarried | 26 (34.7%) |
| | Divorcee | 1 (1.3%) |
| Monthly Income | | |
| | Above minimum wage (≥IDR 4,901,798) | 58 (77.3%) |
| | Below minimum wage (<IDR 4,901,798) | 17 (22.7%) |
| Sufficient for 1 month | | |
| | Sufficient | 67 (89,3%) |
| | Insufficient | 8 (10.7%) |
| Duration of SLE (months) | | |
| | Mean ± SD (Min-Max) | 65.4 ± 64.92 (1-276) |
| Use of steroids | | |
| | Low dose < 7.5 mg/day | 57 (76%) |
| | High dose ≥ 7.5 mg/day | 18 (24%) |
| Sleep Quality (PSQI) | | |
| | Good (PSQI ≤5) | 40 (53.3%) |
| | Poor (PSQI >5) | 35 (46.7%) |
| Quality of Life (Lupus QoL) | | |
| | Mean ± SD (Min-Max) | 84.27 ± 12.59 (39-100) |
| Depression (DASS-21) | | |
| | Normal | 72 (96%) |
| | Depression (Mild-Severe) | 3 (4%) |
| Anxiety (DASS-21) | | |
| | Normal | 72 (96%) |
| | Anxiety (Mild-Severe) | 3 (4%) |
| Disease activity (MEX-SLEDAI) | | |
| | Mild (0–4) | 62 (82.7%) |
| | Moderate (5–9) | 10 (13.3%) |
| | Severe (≥10) | 3 (4%) |
| Pain scale (VAS) | | |
| | Mean ± SD (Min-Max) | 19.61 ± 27.52 (0-100) |
| | Mild (0–39) | 46 (61.3%) |
| | Severe (40–100) | 29 (38.7%) |

**Table 2. Analysis of the association between QoL and SLE-related variables.**

| Variables | | n | Mean ± SD | Median | Min/Max | P value |
|---|---|---|---|---|---|---|
| **Sleep Quality (PSQI)*** | | | | | | **<0.001** |
| | Good | 40 | 88.95 ± 9.47 | 93.46 | 65.33/100 | |
| | Poor | 35 | 78.91 ± 13.67 | 82.42 | 39.9/97.29 | |
| **Age**** | | | | | | **0.014** |
| | Young | 43 | 81.1 ± 12.67 | 80.94 | 49.67/100 | |
| | Middle age | 29 | 88.13 ± 11.67 | 90.21 | 39.9/99.38 | |
| | Older Adults | 3 | 92.39 ± 7.81 | 93.33 | 84.15/99.69 | |
| **Education levels**** | | | | | | 0.050 |
| | Elementary | 7 | 79.84 ± 18.95 | 82.19 | 39.9/95.25 | |
| | Middle | 38 | 81.71 ± 12.42 | 82.74 | 49.67/100 | |
| | High | 30 | 88.54 ± 10.07 | 91.25 | 54.67/99.69 | |
| **Marital status **** | | | | | | 0.129 |
| | Married | 48 | 85 ± 12.99 | 87.07 | 39.9/100 | |
| | Unmarried | 26 | 82.33 ± 11.76 | 80.16 | 63.02/97.19 | |
| | Divorcee | 1 | | | | |
| **Monthly income*** | | | | | | 0.139 |
| | Above minimum wage | 58 | 85.84 ± 10.9 | 87.33 | 49.67/100 | |
| | Below minimum wage | 17 | 78.89 ± 16.46 | 81.85 | 39.9/99.38 | |
| **Sufficiency for 1 month*** | | | | | | 0.482 |
| | Sufficient | 67 | 84.04 ± 12.53 | 86.44 | 39.9/100 | |
| | Insufficient | 8 | 86.17 ± 13.87 | 87.29 | 57.13/99.69 | |
| **Duration of SLE**** | r = 0.167 | | | | | 0.151 |
| **Use of Steroid*** | | | | | | 0.995 |
| | Low dose <7.5 mg/day | 57 | 84.14 ± 12.96 | 86.98 | 39.9/100 | |
| | High dose ≥ 7.5 mg/day | 18 | 84.67 ± 11.69 | 84.96 | 57.13/97.29 | |
| **Anxiety/ Depression (DASS-21)*** | | | | | | **0.006** |
| | Normal | 72 | 85.42 ± 11.4 | 87.07 | 39.9/100 | |
| | Anxiety/Depression (Mild-severe) | 3 | 56.66 ± 8.17 | 54.67 | 49.67/65.65 | |
| **Disease Activity (MEX-SLEDAI)**** | | | | | | 0.297 |
| | Mild | 62 | 83.67 ± 12.54 | 86.09 | 39.9/100 | |
| | Moderate | 10 | 88.1 ± 13.84 | 95.89 | 57.13/99.38 | |
| | Severe | 3 | 83.74 ± 11.18 | 83.06 | 72.92/95.25 | |
| **Pain score (VAS)*** | r = 0.051 | | | | | 0.663 |

Erythematosus (SLE) at Siloam Hospital Lippo Village. Patients with good sleep quality (n = 40) had higher median scores across all domains than those with poor sleep quality (n = 35). Notably, significant associations were observed in physical health (median: 92.5 [52.5/100] vs. 80 [40/100], p = 0.003), pain (100 [53.3/100] vs. 80 [33.3/100], p = 0.003), planning (100 [46.7/100] vs. 93.3 [20/100], p = 0.006), intimate relationships (100 [60/100] vs. 100 [60/100], p = 0.331), burden to others (100 [40/100] vs. 80 [20/100], p = 0.051), emotional health (90 [20/100] vs. 80 [30/100], p = 0.007), body image (96 [52/100] vs. 96 [20/100], p = 0.192), and fatigue (80 [45/100] vs. 65 [20/95], p < 0.0001). The most pronounced difference was in the fatigue domain, where poor sleep quality was strongly associated with lower scores, indicating its significant impact on QoL.

**Table 3. Analysis of the association between sleep quality and each domain of quality of life based on LupusQoL.**

| Lupus QoL Domain | Median (Min/Max) | | P value |
| --- | --- | --- | --- |
| | Good sleep quality<br>n = 40 | Poor sleep quality<br>n= 35 | |
| Physical health | 92.5 (52.5/100) | 80 (40/100) | **0.003** |
| Pain | 100 (53.33/100) | 80 (33.33/100) | **0.003** |
| Planning | 100 (46.67/100) | 93.33 (20/100) | **0.006** |
| Intimate relationship | 100 (60/100) | 100 (60/100) | 0.331 |
| Burden to others | 100 (40/100) | 80 (20/100) | 0.051 |
| Emotional health | 90 (20/100) | 80 (30/100) | **0.007** |
| Body image | 96 (52/100) | 96 (20/100) | 0.192 |
| Fatigue | 80 (45/100) | 65 (20/95) | **<0.001** |

### 3.3 Multivariate analysis of QoL and related variables

As shown in Table 4, multivariate linear regression model was constructed with HRQoL as the dependent variable, independent variables included age, education level, sleep quality (PSQI), and psychological distress (DASS-21), yielding an $R^2$ of 0.361 (p<0.0001), indicating that these factors accounted for 36.1% of the variance in QoL. Depression or Anxiety had the strongest impact, with a β coefficient of −21.402 (p=0.001), suggesting that higher depression or anxiety levels significantly decrease QoL. Poor sleep quality, as measured by the global PSQI score, also significantly reduced QoL, with a β coefficient of −8.392 (p=0.001). Conversely, age was positively associated with QoL (β=5.526, p=0.011), indicating that older age was associated with slightly higher QoL scores.

These findings highlight the critical roles of depression or anxiety and sleep quality as major determinants of QoL in SLE patients, with age playing a less pronounced but still significant role. The multivariate analysis identified 'Anxiety or Depression' as a significant predictor (β=−21.402, p=0.001). However, given that this subgroup comprised only 4% of the cohort (n=3), this coefficient should be interpreted as an indicator of the high magnitude of impact in extreme cases rather than a precise estimate for the broader population. The final multivariate regression model yielded an adjusted $R^2$ of 0.361, indicating that sleep quality, anxiety, and depression explain approximately 36.1% of the variance in HRQoL in this cohort, suggesting that other unmeasured factors (e.g., organ damage or social support) also contribute to the remaining variance.

### Discussion

This study investigated the association between sleep quality and quality of life (QoL) among 75 female patients with Systemic Lupus Erythematosus (SLE) at Siloam Hospital Lippo Village, conducted between February and May 2023. Our study confirms that poor sleep quality is a major detriment to quality of life in women with SLE, underscoring sleep as a critical therapeutic target. The mean Lupus QoL score was 78.91±13.67, indicating a moderate impact on QoL, with fatigue identified as the most significant domain affecting patients (p<0.0001). These results align with prior studies, such

**Table 4. Multivariate analysis of variables and QoL.**

| Variables | β | P value | $R^2$ |
| --- | --- | --- | --- |
| Global PSQI score | −8.392 | 0.001 | 0.361 |
| Age | 5.526 | 0.011 | |
| Anxiety/Depression | −21.402 | 0.001 | |

as Tench et al. [11] and Costa et al. [12], which reported a prevalence of sleep disturbance in SLE patients ranging from 56% to 60%, underscoring the pervasive nature of sleep impairment in this population.

The significant association between sleep quality and QoL underscores the bidirectional relationship between these factors. Poor sleep quality exacerbates fatigue, a hallmark symptom of SLE, which in turn diminishes physical and emotional functioning, as evidenced by the strong correlation with the fatigue domain of the Lupus QoL [13]. This is consistent with literature indicating that sleep disturbances in chronic inflammatory conditions like SLE amplify proinflammatory immune responses [14], further worsening disease symptoms and QoL. The study's findings suggest that addressing sleep quality could be a critical intervention for improving overall well-being in patients with SLE.

Among the variables analyzed, younger age (Mean±SD = 81.1±12.67; p = 0.014) and the presence of depression or anxiety (Mean±SD = 56.66±8.17; p = 0.006) were significantly associated with lower QoL scores. Linear regression analysis further confirmed the substantial impact of depression or anxiety ($\beta$ = −21.402, p = 0.001), sleep quality ($\beta$ = −8.392, p = 0.001), and age ($\beta$ = 5.526, p = 0.011) on QoL, with an $R^2$ of 0.361, indicating that these factors explain approximately 36.1% of the variance in QoL. Although the prevalence of anxiety and depression was low in this cohort (4%), the associated reduction in HRQoL was clinically significant. The high beta coefficient suggests that when psychological morbidity is present in female SLE patients, it may become the dominant driver of poor quality of life, potentially overshadowing traditional clinical markers.

These results corroborate previous research, such as Yilmaz-Oner et al. [15], which found that psychological factors like anxiety and depression are strong predictors of reduced QoL in SLE patients. The influence of younger age may reflect higher expectations for physical and social functioning, which are disrupted by the chronic nature of SLE.

A major study published in Arthritis Research and Therapy found that psychological variables, such as depression and anxiety, were more powerful predictors of health-related quality of life (HRQoL) than disease-specific factors like organ damage or disease activity [2,15]. This suggests that a patient's emotional state regarding their condition has a greater impact on their well-being than their clinical test results. The concept of illness perception partly explains this phenomenon. How a patient understands and perceives their SLE—for instance, viewing it as a constant, uncontrollable threat—can amplify feelings of anxiety. This heightened anxiety, in turn, can lead to maladaptive coping strategies, avoidance of activity, and hypervigilance to symptoms, all of which severely degrade QoL, regardless of the underlying disease activity level [16,17].

Our study found that anxiety is a stronger predictor than sleep quality. While poor sleep is detrimental, it can often be both a symptom and a consequence of anxiety. High levels of anxiety can lead to difficulty falling asleep and frequent awakenings (insomnia). At the same time, the resulting fatigue from poor sleep can exacerbate feelings of anxiety and being overwhelmed during the day [18]. Therefore, anxiety can be seen as a more fundamental driver that impacts QoL both directly and indirectly through its effect on sleep.

The paradoxical association between younger age and poorer QoL likely reflects the 'burden of a life interrupted' during peak productive years [13,19]. In Indonesia, women in their 20s and 30s are navigating critical transitional phases, including the completion of higher education and the initiation of careers. More significantly, cultural expectations surrounding marriage and motherhood—central pillars of social identity in Indonesia—create unique stressors for young SLE patients. Concerns regarding fertility, the ability to manage a household (the 'Ibu Rumah Tangga' role), and the potential stigma of a chronic, visible illness can lead to profound psychological distress and lower HRQoL. In contrast, older patients may possess more mature coping mechanisms or have already achieved many of these life milestones, resulting in a more resilient HRQoL profile despite the chronic nature of the disease [20].

Interestingly, no significant associations were found between QoL and socioeconomic status, disease activity (measured by MEX-SLEDAI), pain scale, or corticosteroid dose. This contrasts with some studies, such as Lau and Mak (2009) [21], which linked socioeconomic factors to QoL. Our findings regarding socioeconomic status should be interpreted within the context of the study setting. Approximately 77.3% of participants reported incomes above the local minimum wage, a

figure higher than the national average for SLE patients in Indonesia. As recruitment took place in a private hospital clinic, this cohort likely represents a segment of the population with better health literacy and more stable financial access to supportive care, such as nutrition and mental health resources. Consequently, the HRQoL challenges faced by the most vulnerable, low-income SLE populations—who may experience even greater disparities due to financial toxicity and limited access to specialized care—may be underrepresented in this study.

The lack of association with corticosteroid use may be due to the study's focus on average prednisone-equivalent doses, which may not capture the nuanced effects of long-term or high-dose regimens on sleep and QoL. Additionally, the relatively small sample size and specific hospital setting may limit the generalizability of these findings. Our cohort predominantly exhibited low disease activity (82.7%), as captured by the MEX-SLEDAI. This suggests that in patients whose systemic inflammation is well-controlled, HRQoL is driven more by lingering symptomatic burdens—such as sleep fragmentation and psychological distress—than by active serological flares. Although the absence of the SLICC/ACR Damage Index (SDI) prevents us from distinguishing between acute activity and permanent physiological damage, the high prevalence of low-activity patients suggests that accumulated organ damage may have been a less prominent driver in this specific outpatient group than in higher-disease-activity populations.

While the current model explains 36.1% of the variance in HRQoL, a substantial proportion (63.9%) remains unattributed. This residual variance likely reflects complex multidimensional factors not captured in this study. In the Indonesian context, social support and cultural stigma may play pivotal roles; patients often face unique psychosocial pressures related to traditional beliefs about autoimmune diseases or family role expectations. Furthermore, permanent organ damage (as measured by the SLICC/ACR Damage Index) was not assessed, which may influence long-term HRQoL independently of the acute disease activity (SLEDAI) measured here. These factors represent critical avenues for future longitudinal research.

Our findings suggest that addressing sleep and psychological distress is paramount for improving HRQoL in SLE. Within the Indonesian healthcare framework, where access to specialized psychiatric care may be limited by resources or stigma, a stepped-care approach is most feasible. For sleep disturbances, Brief Behavioral Treatment for Insomnia (BBTI)—focusing on stimulus control and sleep restriction—offers a streamlined alternative to full-scale Cognitive-Behavioral Therapy (CBT-I) and can be delivered by trained nurses or through digital health platforms, which have seen high uptake in urban Indonesian populations [22]. For anxiety management, integrating psychoeducation and mindfulness-based stress reduction (MBSR) into routine immunology clinic visits is recommended [23]. These 'low-intensity' interventions can mitigate the 'life interrupted' burden in young women by providing accessible coping mechanisms without the logistical hurdles of frequent specialized referrals.

## Strengths and Limitations

The study's strengths include a comprehensive assessment of multiple variables using validated instruments such as the PSQI, Lupus QoL, and DASS-21, providing a robust framework for understanding the interplay among sleep, psychological factors, and QoL in SLE. The results support routine screening for sleep quality and anxiety in patients with SLE, utilizing validated instruments such as the PSQI and DASS-21. Management should adopt a multidisciplinary approach, integrating cognitive-behavioral therapy for insomnia (CBT-I) and anxiety management alongside standard rheumatological care.

A primary limitation of this study is the absence of a healthy control group, which precludes direct comparisons of sleep quality and HRQoL between SLE patients and the general Indonesian population. This absence also limits our ability to draw causal inferences from our findings, as the lack of a control group prevents us from establishing the direction of the observed relationships. Second, generalizability may be limited by the absence of a diverse sample that includes individuals without SLE. However, the within-cohort analysis provides a focused examination of the factors that drive morbidity specifically within SLE. While poor sleep affects the general population, these findings suggest that, in SLE, sleep quality is a critical independent predictor of HRQoL, even after controlling for disease activity (MEX-SLEDAI).

Third, although the prevalence of clinical anxiety and depression in this cohort was low (n = 3), the significant correlation between these psychosocial scores and diminished HRQoL highlights their potential as high-impact areas for clinical intervention. Future longitudinal studies including age- and sex-matched controls are warranted further to isolate lupus-specific sleep pathology from general population trends. Fourth, the low prevalence of clinically significant anxiety and depression (n = 3) limits the generalizability of the regression coefficients for these variables. While the statistical significance remains high, the small subgroup size may lead to an overestimation of the effect size (small-sample bias). Future studies with larger, multicenter cohorts are needed to precisely quantify this relationship.

Furthermore, the current study did not use the SLICC/ACR Damage Index (SDI) to assess permanent organ damage, a recognized driver of long-term HRQoL. The assessment focused primarily on current disease activity, as measured by the Mexican SLE Disease Activity Index (MEX-SLEDAI), and patient-reported symptoms. Given that the cohort predominantly comprised patients with low disease activity (82.7%), the impact of accumulated damage might be less pronounced than in a population with high-acuity systemic involvement. Nevertheless, the absence of SDI scores may limit the ability to distinguish between the effects of acute inflammatory activity and permanent physiological damage on sleep and psychological outcomes. Future research should integrate damage indices to provide a more comprehensive longitudinal perspective on the physical determinants of HRQoL.

Finally, this study focused on clinical and basic demographic predictors. The inability to account for 63.9% of the HRQoL variance suggests that unmeasured variables, such as social support systems, cultural perceptions of chronic illness, and cumulative organ damage, are likely significant contributors to the quality of life in this population. The single-center design at a private hospital may limit the sample's representativeness of the broader Indonesian population, which is diverse in socioeconomic status.

## Conclusion

In conclusion, this study confirms a significant association between poor sleep quality and reduced QoL in female SLE patients, with anxiety, depression, and younger age as key contributing factors. These findings underscore the need for integrated clinical approaches that address sleep disturbances and psychological health alongside traditional SLE management. Clinicians should consider routine screening for sleep disorders and anxiety in SLE patients during regular check-ups. Further, referral to mental health professionals for the management of psychological factors such as anxiety and depression should be a standard practice. Implementing specific interventions, such as cognitive-behavioral therapy for insomnia and anxiety, and providing sleep hygiene education could improve patient outcomes. Future research should employ longitudinal designs and larger, more diverse samples to explore these relationships further and to inform targeted interventions to improve QoL in patients with SLE.

## Supporting information

**S1 Data. SPSS LUPUS QOL.**
(XLSX)

## Author contributions

**Conceptualization:** Euphemia Seto Anggraini Widyastuti, Stevent Sumantri.

**Data curation:** Josephine Lavina, Gabriel Justin Darmajaya.

**Formal analysis:** Josephine Lavina, Stevent Sumantri.

**Investigation:** Euphemia Seto Anggraini Widyastuti, Josephine Lavina, Gabriel Justin Darmajaya.

**Methodology:** Stevent Sumantri.

 

**Project administration:** Euphemia Seto Anggraini Widyastuti.

**Supervision:** Stevent Sumantri.

**Writing – original draft:** Euphemia Seto Anggraini Widyastuti, Josephine Lavina.

**Writing – review & editing:** Stevent Sumantri.

## Acknowledgments

All individuals who made substantial contributions to the work reported in the manuscript (such as technical assistance, writing and editing support, or general support), but who do not meet the criteria for authorship, are named in the Acknowledgements and have provided written permission to be named. If no Acknowledgement is included, this indicates that no substantial contributions were received from non-authors.

Disclaimer:The views represented in this article are the author's point of view and not the institution's official position or funder.

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
