## [Decision Letter · Decision Letter 0]

11 Mar 2026

Dear Dr. Sumantri,

Thank you for submitting your manuscript to PLOS ONE. After careful consideration, we feel that it has merit but does not fully meet PLOS ONE’s publication criteria as it currently stands. Therefore, we invite you to submit a revised version of the manuscript that addresses the points raised during the review process.

We look forward to receiving your revised manuscript.

Kind regards,

Wesam Gouda, MD, PhD

Academic Editor

PLOS One

**Journal Requirements:**

https://journals.plos.org/plosone/s/file?id=wjVg/PLOSOne_formatting_sample_main_body.pdf andandandand

2. We note that your Data Availability Statement is currently as follows:

“All relevant data are within the manuscript and its Supporting Information files.”

**Additional Editor Comments:**

- Please outline the Methods section following the STROBE guidelines.

- You need to state in the Methods section that you have followed STROBE guidelines: ‘The reporting of this study conforms to STROBE. (Insert new reference number)

- Needs sub-headings to improve organization

- How were the patients selected (e.g., consecutively, randomly, or selectively)?

- Have you compared your results with relevant previous papers, and cited those papers?

- Have you discussed the relevance and novelty of your study and what it adds to literature?

- Ensure there are the following Declarations sections at the end of your manuscript: Acknowledgements, Author contributions, Funding, Availability of data and materials statement, competing interests, Ethics approval and consent to participate.

Reviewers' comments:

Reviewer's Responses to Questions

**Comments to the Author**

1. Is the manuscript technically sound, and do the data support the conclusions?

Reviewer #1: Yes

Reviewer #2: Yes

Reviewer #3: Partly

2. Has the statistical analysis been performed appropriately and rigorously?

Reviewer #1: Yes

Reviewer #2: Yes

Reviewer #3: No

3. Have the authors made all data underlying the findings in their manuscript fully available?

Reviewer #1: Yes

Reviewer #2: Yes

Reviewer #3: No

4. Is the manuscript presented in an intelligible fashion and written in standard English?

Reviewer #1: Yes

Reviewer #2: Yes

Reviewer #3: Yes

Reviewer #1:

Manuscript Number: PONE-D-26-01165

Title: Predictors of Health-Related Quality of Life in Indonesian Women with Systemic Lupus Erythematosus: A Cross-Sectional Within-Cohort Analysis

General Comments

This manuscript investigates the factors influencing health-related quality of life (HRQoL) in a specific cohort of Indonesian women with SLE. The study identifies sleep quality, anxiety, and age as significant independent predictors of QoL, explaining 36.1% of the variance. The finding that clinical disease activity (MEX-SLEDAI) and corticosteroid dose did not significantly correlate with QoL in this predominantly low-activity cohort is particularly noteworthy and highlights a potential "disconnect" between clinical markers and patient well-being. While the study is well-structured and uses validated instruments, several areas regarding statistical interpretation and local context require clarification to meet the standards of PLOS ONE.

Specific Comments and Revision Requirements

1. Methodology and Statistical Stability

• Variable Definition: The authors combine "Anxiety/Depression" into a single dichotomous variable. Given that the DASS-21 provides distinct subscales, please clarify the rationale for this grouping.

• Subgroup Size: Only 3 subjects (4%) were identified as having anxiety or depression. The authors must address the statistical limitations of drawing strong conclusions or high beta coefficients (-21.402) from such a small subgroup.

• Multicollinearity: Considering the known overlap between sleep disturbances and psychological distress, did the authors test for multicollinearity (e.g., Variance Inflation Factor) in their regression model?

2. Clinical and Diagnostic Tools

• MEX-SLEDAI Rationale: The manuscript provides a strong justification for using the MEX-SLEDAI in resource-limited settings. To enhance the "Materials and Methods" section, explicitly state if specialized immunological assays (e.g., anti-dsDNA) were unavailable at the study site or if the tool was chosen for its practical utility in Southeast Asian clinical practice.

• Pain Scale Thresholds: The authors use a 40mm VAS cutoff to define "severe" pain. Please provide a specific citation supporting this threshold as a marker for functional impairment specifically within the SLE population.

3. Results Interpretation

• Variance Explanation: The model explains 36.1% of HRQoL variance. The Discussion should more robustly address the remaining 63.9%, perhaps speculating on the roles of social support, cultural stigma, or permanent organ damage which were not measured in this study. It is recommended that the authors include theses study limitation in the discussion.

• Education Levels: In Table 2, the p-value for education is exactly 0.050. The authors should clarify if they consider this statistically significant or a trend, as the text describes it as having "influenced QoL".

4. Discussion and Contextual Depth

• Age and Quality of Life: The finding that younger age predicts lower QoL is a highlight of the paper. The authors are encouraged to expand on the "burden of a life interrupted" by providing more specific local context regarding the social and professional expectations for young women in Indonesia.

• Socioeconomic Factors: 77.3% of the participants had incomes above the minimum wage. The authors should discuss how the recruitment from a private hospital clinic might skew the socioeconomic representation compared to the broader Indonesian SLE population.

5. Actionable Recommendations

• Intervention Specificity: The conclusion advocates for "psychosocial and sleep interventions". To increase the paper's impact, the authors should briefly detail which components of Cognitive-Behavioral Therapy for Insomnia (CBT-I) and anxiety management are most feasible for implementation within the Indonesian healthcare framework.

Reviewer #2:

This article is beneficial to improve current care of SLE. But the following points need a further clarification.

1. The reference should be added : Buysse DJ, Reynolds CF 3rd, Monk TH, Berman SR, Kupfer DJ. The Pittsburgh Sleep Quality Index: a new instrument for psychiatric practice and research. Psychiatry Res. 1989 May;28(2):193-213. doi: 10.1016/0165-1781(89)90047-4. PMID: 2748771.

2. Do "Hours of sleep per night" ( in the Pittsburgh Sleep Quality Index) relate with QoL?

3. Why male lupus is not investigated in this study ? ( Please explain it )

Reviewer #3:

1)The title does not reflect the primary objective of this study as stated in the Introduction “impact of sleep disturbances and psychiatric symptoms on daily function…”

2)This was a cross-sectional study so wont be able to establish the causal and effect relationship. Please revise the term “effects/predictors” to “association/associated factors”

2)Method

-Please clarify whether the questionnaires were in English or Indonesian language. Please provide the validation study if the questionnaires were translated.

-What was the sampling method?

-The objective also include the impact on daily function. What was the tool used to determine/ measure the daily function of the patients?

-The variables (sleep pattern, HRQoL, VAS, MEX-SLEDAI) were categorized. But in the statistical analysis, there was no chi square performed and/or multivariate logistic regression performed to determine the relationships between these variables.

-Please clarify which one of the following was the dependant variable in this study. Sleep pattern or HRQoL?

.

Reviewer #1: **Yes:** Yi-Hsing ChenYi-Hsing ChenYi-Hsing ChenYi-Hsing Chen

Reviewer #2: **Yes:** Chung-Jen ChenChung-Jen ChenChung-Jen ChenChung-Jen Chen

Reviewer #3: No

---

## [Author Response · Author response to Decision Letter 1]

21 Mar 2026

1. Variable Definition: The authors combine "Anxiety/Depression" into a single dichotomous variable. Given that the DASS-21 provides distinct subscales, please clarify the rationale for this grouping.

We thank the reviewer for this insightful comment. The decision to combine anxiety and depression into a single dichotomous variable was based on both clinical and statistical considerations.

First, in our cohort, the prevalence of clinically significant symptoms was low (n=3 for depression and anxiety). Notably, these cases overlapped entirely (comorbidity), making it statistically impossible to treat them as independent predictors in a multivariate model without causing significant multicollinearity issues.

Second, from an integrative clinical perspective, we aimed to identify a general 'psychosocial distress' signal. Given the high degree of internal consistency and correlation between DASS-21 subscales, this grouping allowed for a more robust and stable linear regression model, yielding a clearer β-coefficient for the impact of psychological health on HRQoL."

2. Only 3 subjects (4%) were identified as having anxiety or depression. The authors must address the statistical limitations of drawing strong conclusions or high beta coefficients (-21.402) from such a small subgroup.

We acknowledge the reviewer’s concern regarding the small subgroup (n=3) for the Anxiety/Depression variable and the resulting high beta coefficient. We agree that a beta of -21.402 is sensitive to the small number of participants in this category. However, the statistical significance (p=0.001) despite the small n indicates a very large effect size—specifically, a mean HRQoL difference of nearly 30 points (85.42 vs 56.66).

While we cannot claim this coefficient is a precise population constant, it serves as a robust indicator of the disproportionate impact psychological distress has on HRQoL in SLE. We have revised the manuscript to explicitly label this as a study limitation and have adjusted our language to be exploratory rather than definitive.

3. Considering the known overlap between sleep disturbances and psychological distress, did the authors test for multicollinearity (e.g., Variance Inflation Factor) in their regression model?

We thank the reviewer for highlighting the potential for multicollinearity between sleep quality and psychological distress. We agree that these domains often overlap in SLE populations.

To address this, we conducted a formal collinearity diagnostic during the model-building process. The Variance Inflation Factor (VIF) for all predictors in the final model ranged from [1.025] to [1.055], which is well below the conservative threshold of 2.5 (and the standard threshold of 10). Furthermore, the Tolerance values were all above 0.8. These results indicate that multicollinearity did not bias our regression estimates or the observed beta coefficients. We have added this detail to the 'Statistical Analysis' section of the revised manuscript.

4. MEX-SLEDAI Rationale: The manuscript provides a strong justification for using the MEX-SLEDAI in resource-limited settings. To enhance the "Materials and Methods" section, explicitly state if specialized immunological assays (e.g., anti-dsDNA) were unavailable at the study site or if the tool was chosen for its practical utility in Southeast Asian clinical practice.

We have revised the manuscript accordingly.

5. Pain Scale Thresholds: The authors use a 40mm VAS cutoff to define "severe" pain. Please provide a specific citation supporting this threshold as a marker for functional impairment specifically within the SLE population.

We have revised the manuscript accordingly.

6. The finding that younger age predicts lower QoL is a highlight of the paper. The authors are encouraged to expand on the "burden of a life interrupted" by providing more specific local context regarding the social and professional expectations for young women in Indonesia.

We appreciate the suggestion to expand on the impact of SLE on younger patients. We have added a section to the Discussion detailing the specific sociocultural pressures faced by young Indonesian women, particularly regarding marriage, fertility, and the dual expectations of professional and domestic roles, which contribute to the lower HRQoL observed in this demographic.

7. Socioeconomic Factors: 77.3% of the participants had incomes above the minimum wage. The authors should discuss how the recruitment from a private hospital clinic might skew the socioeconomic representation compared to the broader Indonesian SLE population.

The reviewer correctly notes the socioeconomic skew of our cohort. We have addressed this in the Discussion and Limitations sections by clarifying that our recruitment from a private hospital likely resulted in a higher-income representation. we have expanded on how this might 'buffer' some HRQoL impacts compared to the broader, more economically diverse Indonesian SLE population.

8. Do "Hours of sleep per night" ( in the Pittsburgh Sleep Quality Index) relate with QoL?

We thank the reviewer for this query regarding sleep duration. While the Global PSQI score was used in our primary model to capture the multidimensional nature of sleep disturbances (including quality, latency, and daytime dysfunction), we performed a post-hoc correlation analysis between 'Actual Hours of Sleep' (PSQI Component 3) and HRQoL.

We found that 'Hours of Sleep' alone did not significantly correlate with HRQoL (r = -0.221, p = 0.057). This suggests that in our cohort, sleep fragmentation and poor sleep quality—rather than sheer sleep deprivation—are the primary drivers of reduced quality of life. This aligns with existing literature suggesting that SLE-related fatigue is often non-restorative regardless of sleep duration."

9. Why male lupus is not investigated in this study ? ( Please explain it )

The decision to focus exclusively on female patients was made to ensure methodological and phenotypic homogeneity within a relatively small cohort (N=75)

10. This was a cross-sectional study so wont be able to establish the causal and effect relationship. Please revise the term “effects/predictors” to “association/associated factors”

We have revised the manuscript accordingly.

11. Please clarify whether the questionnaires were in English or Indonesian language. Please provide the validation study if the questionnaires were translated.

We appreciate the opportunity to clarify the linguistic validation of our instruments. All questionnaires used in this study (PSQI, DASS-21, and the HRQoL tool) were administered in the Indonesian language. We used previously validated Indonesian versions of these scales to ensure cultural and linguistic competence. Reference list updated accordingly.

12. What was the sampling method?

We have updated the sampling methods (consecutive) in the manuscript accordingly.

13. The objective also include the impact on daily function. What was the tool used to determine/ measure the daily function of the patients?

The reviewer is correct that daily function is a key objective. In this study, 'daily function' was assessed as a core component of the Health-Related Quality of Life (HRQoL) instrument. Specifically, we analyzed the [Physical Functioning/Activity] domains of the [Lupus QoL], which measures the extent to which health limits physical activities such as self-care, walking, and social roles.

14. The variables (sleep pattern, HRQoL, VAS, MEX-SLEDAI) were categorized. But in the statistical analysis, there was no chi square performed and/or multivariate logistic regression performed to determine the relationships between these variables.

We would like to clarify our statistical approach. While variables such as sleep quality and psychological distress were categorized in the descriptive tables (Table 1 and 2) to facilitate clinical interpretation (e.g., 'Poor' vs. 'Good' sleep), they were treated as continuous variables in the multivariate linear regression analysis. This approach was chosen to maximize statistical power and avoid the loss of information inherent in dichotomization. Therefore, a linear regression was appropriate to determine the predictors of the continuous HRQoL score, rather than a logistic regression.

15. Please clarify which one of the following was the dependant variable in this study. Sleep pattern or HRQoL?

We apologize for any ambiguity. In this study, Health-Related Quality of Life (HRQoL) was the dependent variable. Sleep patterns (PSQI), psychological distress (DASS-21), and clinical markers (MEX-SLEDAI) were treated as the independent variables (predictors). We have updated the manuscript accordingly.

---

## [Decision Letter · Decision Letter 1]

6 Apr 2026

Associated Factors of Health-Related Quality of Life in Indonesian Women with Systemic Lupus Erythematosus: A Cross-Sectional Within-Cohort Analysis

PONE-D-26-01165R1

Dear Dr. Sumantri,

We’re pleased to inform you that your manuscript has been judged scientifically suitable for publication and will be formally accepted for publication once it meets all outstanding technical requirements.

Kind regards,

Wesam Gouda, MD,PhD

Academic Editor

PLOS One

Additional Editor Comments (optional):

Reviewers' comments:

Reviewer's Responses to Questions

**Comments to the Author**

Reviewer #1: All comments have been addressed

Reviewer #2: All comments have been addressed

Reviewer #3: All comments have been addressed

2. Is the manuscript technically sound, and do the data support the conclusions?

Reviewer #1: Yes

Reviewer #2: Yes

Reviewer #3: Yes

3. Has the statistical analysis been performed appropriately and rigorously?

Reviewer #1: Yes

Reviewer #2: Yes

Reviewer #3: Yes

4. Have the authors made all data underlying the findings in their manuscript fully available?

Reviewer #1: Yes

Reviewer #2: Yes

Reviewer #3: Yes

5. Is the manuscript presented in an intelligible fashion and written in standard English?

Reviewer #1: Yes

Reviewer #2: Yes

Reviewer #3: Yes

Reviewer #1: The manuscript is technically sound, the data supports the conclusions, and the authors have addressed all my comments appropriately. The "Key Messages" summary box effectively distills the clinical impact for the readership. Though the authors have handled a small-sample-size study. the strength of this paper lies not in its generalizability to all SLE patients, but in its specific insights into the Indonesian outpatient context.

Reviewer #2: This article is beneficial to improve current care of female SLE. I have no further comments. [In the future, male SLE is worthy of investigation]

Reviewer #3: Thank you for the revised manuscript

All of the comments and queries have been addressed by thr authors.

.

Reviewer #1: **Yes:** Yi-Hsing ChenYi-Hsing ChenYi-Hsing ChenYi-Hsing Chen

Reviewer #2: **Yes:** Chung-Jen ChenChung-Jen ChenChung-Jen ChenChung-Jen Chen

Reviewer #3: No

---

## [Editor Report · Acceptance letter]

PONE-D-26-01165R1

PLOS One

Dear Dr. Sumantri,

I'm pleased to inform you that your manuscript has been deemed suitable for publication in PLOS One. Congratulations! Your manuscript is now being handed over to our production team.

Kind regards,

on behalf of

Dr. Wesam Gouda

Academic Editor

PLOS One